# The Pulp, Peel, Seed, and Food Products of *Persea americana* as Sources of Bioactive Phytochemicals with Cardioprotective Properties: A Review

**DOI:** 10.3390/ijms252413622

**Published:** 2024-12-19

**Authors:** Beata Olas

**Affiliations:** Department of General Biochemistry, Faculty of Biology and Environmental Protection, University of Lodz, Pomorska 141/3, 90-236 Lodz, Poland; beata.olas@biol.uni.lodz.pl; Tel./Fax: +48-42-6354485

**Keywords:** avocado, cardiovascular disease, fatty acid, phenolic compound

## Abstract

Botanically speaking, avocado (*Persea americana*) is a fruit. It consists of a single large seed surrounded by a creamy, smooth-textured edible mesocarp or pulp covered by a thick, bumpy skin. Avocado is a nutrient-dense fruit, containing a range of bioactive compounds which have been independently associated with cardiovascular health. These compounds have been obtained from the pulp, peel, and seed. This narrative review summarizes the current understanding of the cardioprotective potential of avocado fruit, especially the pulp and seed, and its food products, and examines the biological mechanism behind it.

## 1. Introduction

*Persea americana*, known as avocado, alligator pear, avocado pear, ahuacate, or butter pear, is a tropical fruit native to the Americas. It is a member of the genus *Persea*, family *Lauraceae*, which comprises about 50 species. Avocado grows in a warm temperate climate and is believed to have been first cultivated in Mexico as early as 500 BC, although the first mention of the name avocado in the English language is believed to have been in 1969. Currently, it is cultivated in subtropical and tropical parts of the world. Its different varieties are classified into three groups named for the geographic region in which they were domesticated: West Indian (*P. americana* var. *americana*), Guatemalan (*P. americana* var. *guatemalensis*, L.O. Williams), and Mexican (*P. americana* var. *drymifolia*). Each group has unique characteristics, including peel texture, color, and fruit size. For example, the fruit of the Mexican avocado is smaller and contains a higher percentage of oil than the West Indian variety.

The three varieties share a similar genome (2n = 24), and hybridization occurs easily; this facilitated the generation of more than 800 avocado cultivars. The most widely consumed avocado cultivar worldwide is the Hass avocado (*Persea americana*), a Guatemalan/Mexican hybrid responsible for 95% of the total commercialized volume in the United States. Its biological properties are also more widely studied than those of other varieties [1,2,3].

The trees are tropical evergreens which grow to about 20 m, with thick grayish-brown bark and broad leaves. The leaves are 7–41 cm in length. The flowers are yellow or green in color, measuring about 1–1.3 cm in width. The fruit itself is a berry with a single large seed, and its size depends on the cultivar [4,5,6].

The leading producer of avocados is Mexico, responsible for about 30% of global avocado production (about 1,800,000 tons/year in 2019). Other significant producers include the Dominican Republic, with about 9% of the global production, Colombia and Peru, with about 7%, Indonesia, about 6%, Kenya with about 5%, and Brazil, responsible for about 3% of global avocado production. Recent years have seen an exponential increase in avocado consumption in the US, growing from 2.23 pounds per capita in 2000 to 7.1 in 2016, with the most common varieties being “Fuerte”, “Bacon”, “Gwen”, “Lamb Hass”, “Reed”, “Pinkerton”, and “Zutano” [4].

*P. americana* consists of a creamy smooth-textured edible part known as the pulp or mesocarp, covered by a thick dark green, purplish black, and bumpy skin. The entire fruit of *P. americana* (flesh, peel, and seed) weighs between 100 and 1000 g. Its seed averages 5 cm to 6.4 cm in length and constitutes approximately 15–25% of the total weight [5,7]. Dreher and Davenport [7] report that the seed and skin together make up about 33% of the total weight. Avocado pulp contains about 60% oil, which is also used as a food product; although there are no accurate statistics on total avocado oil production, the main producers are New Zealand, the United States, Mexico, Chile, and South Africa [8].

Production and consumption of avocado generate large quantities of by-products, including peel, seeds, and defatted pulp, which account for approximately 30–45% of fruit weight, but, for example, the seed may be used in the food and cosmetic industries: it can be found in flour, orange dye, skin products, and soap [6].

Avocados are a form-to-market food. They require no taste enhancers, preservatives or processing. Moreover, its natural skin eliminates the need for packaging and offers some disease and insect resistance [7]. Fresh avocados are eaten when ripe, and are often consumed in spreads and dips, such as guacamole; they are also eaten on their own, and as toppings for burgers, sandwiches, soups, and salads [3].

Recent studies considerably broadened our understanding of the cardioprotective potential of avocado and its biological properties in general. In addition to its taste, avocado is also consumed for its anti-lithiasis, anticonvulsant, antimicrobial, chemopreventive, and hepatoprotective properties [1,2,9]. Most significantly, avocado and its products are known to improve various cardiometabolic parameters, such as hyperlipidemia, inflammation, insulin and blood glucose concentration, as well as metabolic syndrome [10,11,12]. However, no current comprehensive reviews have yet examined the mechanisms behind these cardioprotective properties. The present work reviews the up-to-date literature concerning the cardioprotective potential of various parts of avocado fruit (*P. americana*), especially its pulp and seed and its food products; it also examines its chemical composition and the mechanisms behind its cardioprotective properties. This review paper is based on a corpus of papers identified in various electronic databases, including PubMed, Science Direct, Scopus, Web of Knowledge, and Google Scholar. No time criteria were applied to the search, but recent papers were evaluated first. The last search was run on 15 November 2024. The databases were searched using the major keywords: “avocado”, “avocado oil”, “*Persea americana*”, “*P. americana*”, “bioactive compound”, “antioxidant”, “cardioprotective”, “anti-platelet”, “anti-inflammatory”, or a combination of these terms. The abstracts of any identified articles were initially analyzed to confirm whether they met the inclusion criteria. Any relevant identified articles were summarized.

## 2. Phytochemical Characteristics of Various Parts of *P. americana*

*P. americana* has a unique nutrient profile characterized by a low-medium energy density, with approximately 72% of its weight being water, and a low sugar content (0.2 g of sugar/half of the fruit). In addition, avocado possesses high palatability and sensory quality. One *P. americana* provides about 250 kcal, 2.7 g protein (4% energy), 11.8 carbohydrate (19% energy), and 21 g fat (76% energy). Dos Santos Tramontin et al. [13] note that avocado is rich in fat (about 15% by wt.), particularly monounsaturated fatty acids (MUFAs) (9.6% by wt.), which constitute 62.8 to 63.6% of total fatty acid content, with a predominance of oleic acid. The ratio of unsaturated to saturated fatty acids is approximately 6:1. In addition, avocado has a high unsaturated fatty acid content compared with other consumed fat sources, i.e., the UFA content of avocado is 85%, compared to 33% for mozzarella cheese and 32% for butter [3]. Additionally, a medium-sized avocado fruit (136 g) contains about 13 g of oleic acid, which is about the same as in two tablespoons (26 g) of olive oil or 1.5 oz (42 g) of almonds [14].

Various studies [3,7,13] examined the fat content of avocado. Avocado oil itself consists of about 71% MUFAs, 13% polyunsaturated fatty acids (PUFAs), and 16% saturated fatty acids (SFAs); these not only have pro-health properties, but also enhance the bioavailability of fat-soluble vitamins (A, D, E, and K) in the avocado or other fruits and vegetables, which may be naturally low in fat.

Avocado carbohydrates consist of about 80% dietary fiber (70% insoluble and 30% soluble dietary fibers), comprising a mix of cellulose, hemicellulose, and pectin. In addition, compared with other fruits and vegetables, avocado has a high fiber content: 6.9 g per 100 g compared to 0.9 g for grapes, 2.4 g for oranges and apples, 2.6 g for bananas, 1.2 g for tomatoes, and 2.8 g for carrots [3]. The pulp has also been found to contain unique C7 carbohydrates [3].

One medium avocado fruit also contains about 690 mg potassium and 39 mg magnesium, which are linked to cardiovascular health. Clinical research indicates that potassium promotes blood pressure control, and magnesium intake is associated with a decreased risk of cardiovascular diseases (CVDs), such as fatal ischemia heart disease [10,11,12].

Moreover, Dreher and Davenport [7] report that avocado also includes various vitamins, including vitamin C (6.0 mg/100 g), vitamin A (43 µg/100 g), vitamin E (1.3 mg/100 g), folate (60 mg/100 g), and phytosterols (57 mg/100 g). Six phytosterols have been detected in avocado pulp: β-sitosterol, stigmasterol, campesterol, cyclosterol, avenasterol, and stanol; the most abundant of these is β-sitosterol (57–76 mg/100 g) [3]. Avocado also contains phytostanols, the main one being cycloarthenol. Avocado oil is also the source of phytosterols and phytostanols such as lanosterol, campestanol, Δ5-avenasterol, Δ7-sitosterol, cycloeucalenol, citrostadienol, and 24-methylenecycloartanol [15].

In addition, one avocado fruit (136 g) has a similar phytochemical profile and nutrient content as 1.5 ounces (42.5 g) of nuts (walnuts, almonds, and pistachios) [7]. Avocado pulp has also been found to contain glutathione (27.7 g/100 g) and betaine (0.7 g/100 g), and is one of the best sources of glutathione among fruits and vegetables [3].

In addition to avocado pulp, the “waste products”, viz. leaf, seed, and peel, also contain a range of bioactive compounds, including phenolic compounds and organic acids, with various lipid-lowering, gastroprotective, hepatoprotective, and antioxidant activities, as well as hypoglycemic properties. Recently, Collignon et al. [11] found that the different parts of *P. americana* contain numerous phenolic compounds, fatty alcohols, terpenoids, and furan derivatives. The fatty alcohols were isolated from unripe fruits, leaves, and stem bark of avocado. Among these fatty alcohols, persenone A, persenone B, and acetylated avocadene have the highest concentrations in avocado pulp [3]. The most abundant is persenone A, representing 48% of total acetogenins in the pulp and 46% in the peel. The bioavailability of avocado acetogenins is unknown, but various studies indicate these compounds exert various biological activities; for example, they appear to exert antioxidant activity by inhibiting the production of nitric oxide and superoxide in cells [3].

Moreover, the pulp also contains the polyhydroxylated alcohols avocadene and avocadyne, unique C7 carbohydrates such as mannoheptulose, perseitol, and volemitol, as well as various phenolic compounds: the total phenolic content is about 20 mg/100 g gallic acid equivalent (GAE) [3]. Dimethyl sciadinonate (terpenoid) has also been described in leaves of *P. americana* [16].

Results of Santana et al. [17] indicate that avocado peel contains flavonoids, phenolic acids including vanillic acid, ferulic acid, and gallic acid, as well as procyanidins and catechins; however, their content depends on the extraction methods [18]. Avocado also possesses high levels of flavonoids in its pulp [19,20]; the main ones in the peel are hesperidin (0.01 mg/g) and epicatechin (0.02 mg/g) [21]. The total phenolic compound content in avocado pulp has been found to range from 1 to 26 mg/100 g [3].

Araujo et al. [22] identified 15 phenolic compounds in avocado peel: 11 procyanidins and various forms of isomers including dimers and trimers. Avocado seed is also a source of flavonoids (e.g., catechin; 1.02 mg/g dry weight) and phenolic acids, including ferulic acid, caffeic acid, and chlorogenic acid [15,23,24]. In addition, various proanthocyanidins have been isolated from the seeds [6]. More details about substances isolated from various *Persea species*, including *P. americana,* are given by Ford et al. [3] and Totini et al. [16].

Avocado oil is mainly obtained from the pulp by mechanical techniques such as cold pressing or chemical solvation with organic solvents. Avocado oil contains high levels of MUFA, constituting 37–86% of total fatty acid content, with the main form being oleic acid; however, there are no internationally defined parameters for avocado oil content. Various authors noted that oil quality and yield depend on the cultivar, quality, and maturity of the fruits, as well as the extraction method and conditions, such as the chosen solvent, temperature, pH, and added enzymes [8]. For example, the quality parameters, including amount of oleic acid, content of carotenoids, phenolic compounds, and antioxidant properties, have shown better results when the pulp is dried at 60 °C under vacuum, and the extraction is performed by the Soxhlet method. However, the bioactive compounds were best preserved when the avocado pulp was dried at 60 °C with air ventilation and mechanical pressing [8].

The resulting oil is used both in cosmetics and food production [25]. Extra-virgin avocado oil, for example, is a viscous edible oil with a dark green color due to its carotenoid and chlorophyll contents. It has a mild taste and it is typically extracted from avocado fruit by cold pressing [25].

Avocado oil may also be extracted from the seed and skin. The edible oil obtained from avocado seeds has its own flavor, color, and texture, and interestingly, crude avocado oil has a similar viscosity to extra-virgin avocado oil. This may not be surprising considering that refined avocado oil is obtained mainly from crude oil. Crude oil is light in color, odorless, free of waxes, and low in free fatty acids [25]. A more detailed physiochemical characterization of the oils from different varieties and origin of avocado is given in a review by Flores et al. [8], who also note that pulp oil is has higher proportions of MUFA than the seed oil, while the seed oil has higher levels of PUFA than the pulp oil [26].

As mentioned above, the carotenoid and chlorophyll content of avocado fruit confer a characteristic dark green appearance to avocado oil [3]. The most abundant carotenoid in avocado oil is lutein, although various other carotenoids, including β-carotene, neoxanthis, zeaxanthin, and violaxanthin are also present [6,25]. For example, oil from avocado skin exhibits higher carotenoid levels than from yellow pulp, being pale and dark. Moreover, the carotenoid and chlorophyll content in the skin of the green fruit decreases as the fruit ripens. Virgin avocado oil should contain high levels of chlorophyll, giving it an emerald green color; in contrast, virgin olive oil has a comparatively low level [27,28]. Krumreich et al. [29] propose that the moderate temperature (60 °C) during the oil extraction enhances its release from cell tissues, while preventing degradation.

Tocopherols have been found to be present at high concentrations, i.e., from 0.034 up to 256 µg/g oil, in avocado oil. The most prevalent forms are α-tocopherol, β-tocopherol, and γ-tocopherol [6,25]. However, industrial fruit processing can result in loss of α-tocopherol and other tocopherols. In addition, the tocopherol content of an avocado oil may also be influenced by its variety [25]. The pro-health potential and uses of avocado fruit and its by-products, and their value as potential sources of bioactive compounds, are given in Figure 1.

Selected metabolites isolated from different part of *P. americana* are presented in Table 1.

## 3. Bioavailability of Avocado Chemical Compounds

Although limited research has been performed on the bioavailability of the compounds in avocado alone, others examined their absorption when used in combination with other foods and supplements [4,44]. Interestingly, avocado can act as a “nutrient booster” by improving the absorption of fat-soluble nutrients found in foods. For example, avocado has been found to improve carotenoid absorption consumed with carotene-rich tomato source or carrots in healthy subjects [45].

Elsewhere, a study of 11 healthy adults found the absorption of β-carotene and lycopene from salsa to increase when avocado (24 g) was added (150 g) [44]. The authors report higher circulating levels of β-carotene (2.6 x) and lycopene (4.4 x) in the salsa with the avocado group compared to the without avocado group; they attribute this process to the high MUFA content of avocado.

Furthermore, the highly bioavailable lutein from avocado also helps protect low-density lipoprotein (LDL) cholesterol from oxidation and can inhibit atherosclerosis, which can benefit cardiovascular health. A higher lutein intake is also associated with lowered risk of stroke and coronary heart disease [46,47].

## 4. Safety of Various Parts of Avocado

Various in vitro studies found different parts of avocado to be safe [48,49,50,51,52]. However, few studies evaluated the safety of various parts of avocado towards human cells (Table 2).

Several reports, however, examined their toxicity towards animals, especially in breast, heart, liver, kidney, and lung tissues. For example, Grant et al. [53] report that avocado leaves (25 g/kg/day, for 32 days) promote congestion of the myocardium of goats, kidney infraction, and can cause severe liver congestion. In addition, Oelrichs et al. [54] found that persin isolated from avocado leaves (60–100 mg/kg) has cytotoxic properties on the acinar epithelium of mammary glands of mice, and inhibits lactation in mice (100 mg/kg). Nevertheless, avocado and its products are usually safe for humans, with toxicity rarely demonstrated at dietary levels. However, further studies are needed before making safety recommendations for avocado consumption in humans.

## 5. Cardioprotective Compounds in Avocado and Its Food Products

The nutrients and bioactive compounds found in the various parts of avocado and its food products, such as MUFAs, PUFAs, phenolic compounds, carotenoids, and tocopherols, are believed to confer several health benefits, including improved cardiovascular health. For example, eating one or more avocados weekly has been associated with a 21% lower risk of coronary heart disease and a 16% lower risk of CVDs [14]. In addition, avocado consumers demonstrate about 30% lower blood platelet aggregation compared to non-consumers [55].

A key factor involved in the development and progression of CVDs is oxidative stress [56]. Studies based on FRAP, oxygen radical absorbance capacity (ORAC), DPPH, and ABTS testing found avocado pulp, leaves, peel, and seeds to have antioxidant activity, and suggest that the key antioxidants are phenolic compounds, tocopherols, carotenoids, MUFA, and PUFA [57,58,59,60]. The seed, peel, and leaf extracts demonstrated greater antioxidant properties than pulp extracts [57,61,62,63,64,65].

Avocado oil also contains a wide variety of lipophilic antioxidants, which may alleviate oxidative stress by decreasing ROS production. It has been noted that avocado oil has various carotenoids and β-sitosterol, which protect against oxidative stress, which is measured by protein carbonylation [66,67,68]. For example, supplementation with β-sitosterol increased the activity of different antioxidant enzymes in mice [68]. Zavala-Guerrero et al. [69] also found avocado seed oil obtained with hexane by the Soxhlet method to have antioxidant potential in vitro based on three radical scavenging methods: total antioxidant capacity (TAC), DPPH, and ABTS. The oil demonstrated strong antioxidant activity. Other studies also found avocado oil to have anti-inflammatory activity [2].

Avocado consumers have been found to have higher intakes of dietary fiber, PUFA, MUFA, potassium, magnesium, and vitamin E, and lower intakes of unhealthy “discretionary” foods; they also have lower body mass index (BMI) than non-consumers [70]. However, these findings are controversial [11,71,72]. For example, body weight reduction was observed in rats fed a sucrose-rich diet and supplemented with 10–30% avocado oil for eight weeks [73]. Supplementation with hydroalcoholic extract from avocado (100 mg/kg body weight, for 12 weeks) also decreased weight gain (24.8%) and BMI (17.9%) in rats fed a high-fat diet [74]. In addition, treatment with 10 mg/kg methanolic avocado leaf extract for eight weeks reduced body weight gain by 25% in hypercholesterolemic rats compared with the control [75]. Recently, Lichtenstein et al. [76] reported similar modifications in body weight, BMI, and very low-density lipoprotein (VLDL) cholesterol concentration between adults who consumed avocado and adults who consumed a habitual diet. These authors also noted significant reduction in total cholesterol and LDL-cholesterol in adults who consumed avocado versus those who did not.

A comprehensive review of Dreher et al. [71], including nineteen trials and five observational studies with avocado oil, demonstrated a reduced risk of CVD in healthy overweight and obese adults with dyslipidemia by increasing peripheral blood flow and high-density lipoprotein (HDL) cholesterol level, decreasing total cholesterol, LDL-cholesterol and triglyceride, and control weight management. The authors also attribute the cardioprotective actions of avocado and its food products to its fiber, phytosterol, phenolic compound, and a 6:1 unsaturated to saturated fat ratio; similar to olive oil, the key unsaturated fatty acid was oleic acid. In addition, avocado and its food products are good sources of compounds with known cardioprotective properties, such as potassium, magnesium, and vitamins such as vitamin A [3,17,38].

About 10% of avocado consists of MUFA, which is a highly recommended alternative to trans fatty acids and saturated fatty acids for the prevention of CVDs. For example, higher MUFA intake has been associated with a reduction in cardiovascular events (9%), all-cause mortality (11%), cardiovascular mortality (12%), and stroke (10%) compared to lower MUFA intake [12]. Pacheco et al. [14] report higher avocado intake to be associated with significantly lower risk of coronary heart disease (CHD, 21% lower risk), and CVD (16% lower risk) in both men (n = 41,701) and women (n = 68,786); the authors also note that replacing yogurt, butter, margarine, cheese, and processed meat with avocado was associated with a lower incidence of CVD events.

A systematic review by Silva-Caldas et al. [77] concluded that the cardioprotective action of avocado is linked with its MUFA content, especially oleic acid. Oleic acid can decrease LDL-cholesterol by increasing acyl-CoA, cholesterol acyltransferase (a liver enzyme that catalyzes the formation of cholesterol esters from cholesterol), activity, thus increasing the synthesis of cholesterol esters; these stimulate the action of LDL receptors, favoring LDL uptake and reducing its presence in plasma [77]. Other studies showed that decreasing plasma triglyceride content and increasing that of HDL-cholesterol may favor the lipase hydrolysis of long-chain fatty acids in the triglycerides, which are incorporated into HDL particles [78]. In addition, oleic acid may reduce endogenous total cholesterol synthesis, and decrease LDL-cholesterol and total cholesterol levels [77].

Scoditti et al. [79] suggest that oleic acid may reduce the risk of CVDs by decreasing the expression of interleukin-6 (IL-6) and tumor necrosis factor α (TNF-α). Other studies suggest that it may regulate the structure of membrane lipids and reduce blood pressure by controlling G protein-mediated signaling. Moreover, Tan et al. [28] noted that the ingestion of avocado oil and simvastatin causes a decrease in the atherogenic index, thus reducing the risk of atherosclerosis. These authors studied the hypocholesterolaemic effect of virgin avocado oil (400 and 900 mg/kg body weight) using diet-induced hypercholesterolaemia rats. Rats were fed a high-cholesterol diet for 4 weeks to induce hypercholesterolaemia. At the end of the experiment, the LDL-cholesterol and triglyceride levels were significantly reduced, while the HDL-cholesterol level was significantly increased in high-dose avocado oil (900 mg/kg body weight per day) when compared with their respective baseline values.

Pahua-Ramos et al. [80] noted that carbohydrates found in avocado may regulate blood cholesterol. However, De la Torre-Carbot et al. [81] propose that the cardioprotective action of avocado may be influenced by the high level of tocopherols in its oil, as observed in rats fed diets with avocado oil.

In addition, this cardioprotective function may be associated with the presence of other phytochemicals, including phenolic compounds; these have all been found to reduce oxidative stress, inflammation, or blood platelet hyperactivation, and thus CVD risk. For example, Wang et al. [57] indicate that these compounds demonstrate antioxidant properties and have a strong inhibitory effect on the activity of COX-1 and COX-2. Data from pre-clinical studies also suggest that phenolic compounds, including the epicatechin found in avocado pulp, have cardioprotective effects [82].

The phytosterols in avocado pulp may have also cardioprotective action and reduce the risk of coronary heart disease. Indeed, they are well known to lower LDL-cholesterol concentrations by inhibiting intestinal cholesterol absorption and decreasing hepatic synthesis [83].

Fatty alcohols also demonstrated anti-platelet activity in vitro and in vivo [15,55]. For example, persenone A (25 mg/kg body weight) attenuated the formation of thrombi; itprevented arterial thrombosis and increased coagulation time [55]. Similarly, persensone A and B, have also exhibited antioxidant properties in vitro by inhibiting superoxide and nitric oxide production [35]. Persenone A has also been found to inhibit the activity of COX-2 and inducible nitric oxide synthase (iNOS) in murine macrophages [35]; similarly, persenone B isolated from avocado fruit also decreased nitric oxide (NO) synthesis in the same cell line [84].

Avocado fruit oil also decreased inflammation in Wistar rats, inhibiting COX-1 and COX-2 in a similar way to ibuprofen and extra virgin olive oil [85]. Marquez-Ramirez et al. [86] report that avocado oil (1 mL of oil per 250 g of fat weight) appears to have antihypertensive action in rats, with avocado oil lowering both diastolic and systolic pressure in hypertensive rats, but not in control. Other studies indicate that the addition of avocado oil in the diet decreased the concentrations of LDL, VLDL, and triglyceride, without affecting the levels of HDL in rats fed with sucrose. This oil also decreased the level of C-reactive protein (CRP), an indicator of inflammation [87].

Avocado intake also improves lipid profiles, i.e., reducing total cholesterol and LDL-cholesterol levels, in adults with dyslipidemia [12,70,71]. However, other research did not find that an avocado diet significantly impacted triglyceride level [12]. Similarly, Mahmassani et al. [10] report that avocado intake results in no difference in serum LDL-cholesterol, total cholesterol, and triglyceride concentrations, and that it increases serum HDL-cholesterol. The authors attribute this effect on HDL-cholesterol level to the increased intake of MUFA, sterols, or fiber from the avocado. Interestingly, substantial heterogeneity was noted between studies in terms of included subjects, control groups, background diet composition, and duration of intervention; for example, four studies included subjects aged under 25 years, whereas others included those aged over 40 years, and while some studies included only healthy subjects, others included subjects with dyslipidemia.

Various studies suggest that avocado can support weight control. Dreher et al. [71] attribute this to reducing hunger and increasing meal satisfaction and satiety. In addition, it has been found that consuming one avocado per day over four to five weeks improves blood lipid profiles in subjects with overweight or obesity [71]. Furthermore, a study of 61 healthy free-living, overweight, and obese subjects consuming one and a half avocados (200 g) per day for six weeks found all groups to demonstrate a decrease in BMI [88].

One systematic review and meta-analysis found supplementation with Mexican ancestral foods, including avocado, to significantly improve BMI in obese patients [72]. However, another systematic review and meta-analysis of randomized controlled trials showed that avocado consumption (136 g/day to 200 g/day) does not improve weight loss in adults with excess weight (BMI ≥ 25 kg/m^2^) [11]. Another recent systematic review and meta-analysis found that avocado consumption does not negatively impact body weight [70].

Interestingly, avocado waste may be an important source of bioactive compounds that can exert beneficial health effects, including cardiovascular health. However, these effects have only been noted on dyslipidemia parameters in animal models [18,75,80,81,82,83,84,85,86,87,88,89,90,91]. More details about it were described by Pineda-Lozano et al. [18]. A recent review by Charles et al. [5] proposes various food applications for avocado seeds, such as avocado seed flour, but does not discuss the health effects of avocado seeds.

The cardioprotective effects of various parts of avocado and its food products, especially the oil and its active constituents, based on in vitro, animal, and clinical models are summarized in Table 3; it can be seen that the cardioprotective activity of avocado and its products varies according to a number of factors, including experimental method, dosage, and route of administration.

The cardioprotective potential of avocado and its food products is also summarized in Figure 2, together with their biological mechanism. It appears that avocado possesses antioxidant potential, manifested by inhibiting ROS generation and stimulating the activity of antioxidant enzymes. It also exerts cardioprotective activity, which seems also to be associated with its anti-inflammatory properties, i.e., by inhibiting COX, IL-6 and TNF-α expression, and reducing NO production by iNOS

## 6. Conclusions

The effect of avocado and its oil on cardiovascular health has been evaluated in a range of animal and human models. The action is generally attributed to the fatty acid content of the fruit. However, little is known of the effect of avocado waste, although studies suggest it may have pro-health potential. In addition, the effects of fiber and phytochemicals such as phenolic compounds, carotenoids, and tocopherols on cardiovascular health received limited attention; these merit further research, as they may have better preventative activity against CVDs than fatty acids. In addition, longer and larger clinical trials are needed to better understand the cardioprotective properties of avocado and its food products.

## Figures and Tables

**Figure 1 ijms-25-13622-f001:**
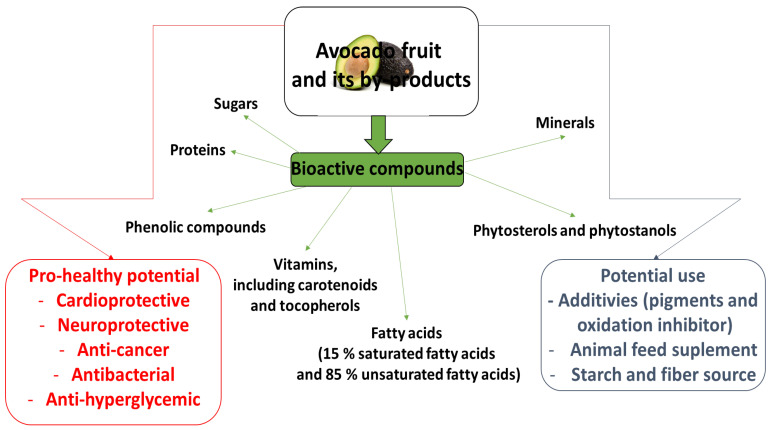
Avocado fruit and its by-products as potential source of bioactive compounds, their pro-healthy potential and use [19] (modified).

**Figure 2 ijms-25-13622-f002:**
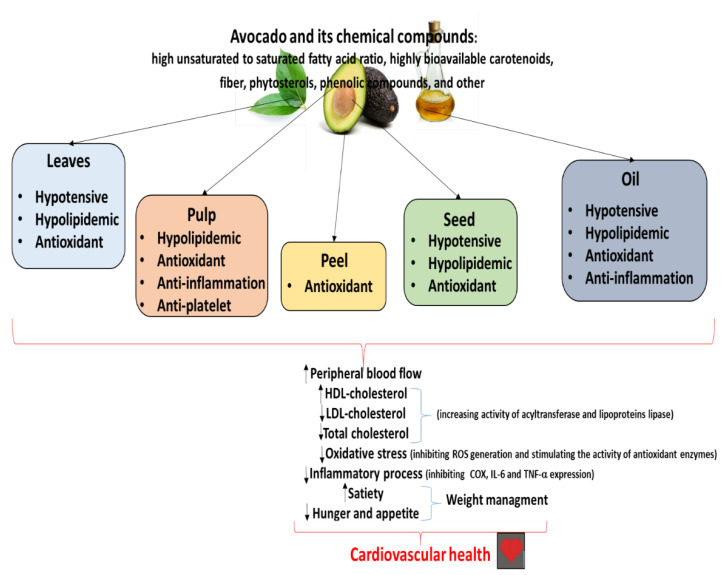
Biological mechanisms of cardioprotective properties of avocado and its chemical compounds [71] (modified).

**Table 1 ijms-25-13622-t001:** Selected metabolites isolated from different parts of *P. americana* and their biological activity.

Part of *P. americana*	Chemical Compound	Biological Activity	References
**Fatty alcohols**			
Pulp and seeds	Avocadyne, avocadenol D	Antifungal, insecticidal, inhibition of virus replication, and cytotoxic	[30,31,32]
Peel and leaves	Avocadoin	Antifungal, insecticidal, and cytotoxic	[30,33,34]
Pulp	Persenone A (0.2–4.6 g/kg FW) and B	Antioxidant	[35]
Bark	Secosubamolide	Cytotoxic	[36]
**Phenolic compounds**			
Pulp oil	Quercetin, gallic acid, vanillic acid, coumaric acid, and ferulic acid	Antioxidant	[17]
Seeds	Proanthocyanidins B1, B2 and A-type trimer, catechin, ferulic acid, caffeic acid, and chlorogenic acid	Cytotoxic and antioxidant	[6,15,23,24,37]
Peel	Vanillic acid, ferulic acid, gallic acid, hesperidin, procyanidins, dimers, and trimers in various shapes	Antioxidant	[3,17,18,22]
**Carotenoids**			
Pulp and pulp oil	α-carotene (pulp—24 µg/100 g), β-carotene (pulp—63 µg/100 g), zeaxanthin, and β-cryptoxbthin (pulp—0.0002–0.0003 g/kgFW)	Antioxidant and cytotoxic	[3,17,38]
Pulp	α-citraurin	Nutrient, energy source and storage	[39]
**Carbohydrates**			
Pulp	D-manno-2-heptulose (9–348 mg/100 g), D-erythro-L-gluco-Nonulose, D-erythro-L-galacto-Nonulose	Nutrient, energy source and storage	[40,41,42]
**Furan derivatives**			
Seed and pulp	Avocadienofuran	No information	[43]
**Tocopherols**			
Pulp and pulp oil	α-tocopherol (pulp—1.97 mg/100 g) and β-tocopherol (pulp—0.04 mg/100 g)	Antioxidant	[17]

**Table 2 ijms-25-13622-t002:** Effect of different parts of avocado on toxicity of cells in vitro.

Different Parts of Avocado and Its Bioactive Compounds	Concentration and Incubation Time	Type of Cells	Toxic Effect	References
Avocado peel extract	8–73 µg/mL, 24 h	L929 normal mouse fibroblast	No toxic effect	[51]
Avocado peel extract	100 µg/mL, 24 h	RAW 264.7 macrophage cells	No toxic effect	[48]
Avocado peel fraction	226 µg/mL, 24 h	African green monkey kidney cells	No toxic effect	[52]
Procyanidins from avocado seeds and peels	1–50 µg/mL, 18 h	HEK293 human embryonic kidney cells	No toxic effect	[50]
Lipid extract of avocado seeds	150 µg, 48 h	Human erythrocytes	No toxic effect	[49]

**Table 3 ijms-25-13622-t003:** The cardioprotective effects of various parts of avocado, its food products, and its active constituents in different experimental models (animal models and clinical models).

Part of Avocado/Its Food Product	Active Constituents	Dosage	Experimental Model	Cardioprotective Action	References
Animal model					
Pulp	Phenolic compounds, tocopherols, and phytosterols	130 and 150 mg/kg/day, 8 weeks	Rats with high-cholesterol diet	Hypolipidemic action	[92]
Pulp	MUFAs, phytosterols, and carotenoids	1 and 2 mL/rat/day, 10 weeks	Rats with high-cholesterol diet	Hypolipidemic action	[93]
Pulp	Persenone A	25 mg/kg of body weight	Mice	Antithrombotic action	[55]
Fruit	MUFAs	25–200/p.o., 14 weeks	Obesity rats with high-fat diet	Hypolipidemic action	[94]
Fruit	Phenolic compounds, and phytosterols	100/p.o., 14 weeks	Rats with high-fat diet	Hypolipidemic action	[74]
Fruit	Phenolic compounds, and phytosterols	100/p.o., 14 weeks	Rats with high-fat diet	Hypolipidemic action	[95]
Fruit and seed	Phenolic compounds, and fiber	10–30%/p.o., 4 weeks	Rats with high-cholesterol diet	Hypolipidemic action	[96]
Seed	Phenolic compounds	125–500/p.o., 4 weeks	Rats with high-fat diet	Hypolipidemic action	[97]
Seed	Phenolic compounds	10–40/p.o., 2 weeks	Hyperlipidemic rats	Hypolipidemic action	[98]
Seed flour	Phenolic compounds, and fiber	125–500/Gavage, 2 weeks	Rats with high-cholesterol diet	Hypolipidemic action	[80]
Leaves	Phenolic compounds	20 and 40/p.o, 8 weeks	Rats with high-cholesterol diet	Hypolipidemic action	[99]
Leaves	MUFAs	10/p.o., 8 weeks	Rats with high-fat diet	Anti-obesity action	[100]
Leaves	Phenolic compounds	10/p.o., 8 weeks	Rats with high-fat diet	Antioxidant action	[75]
Paste	Phenolic compounds, carotenoids, and fiber	2000/p.o., 7 weeks	Rats with high-cholesterol diet	Hypolipidemic action	[89]
Oil	PUFAs, carotenoids, and phytosterols	No data, 3 months	Atherogenic rabbits with high-fat diet	Hypolipidemic action	[101]
Oil	MUFAs, carotenoids, and phytosterols	2.5 or 5%, 4 weeks	Mice with high cholesterol diet	Hypolipidemic action	[102]
Oil	MUFAs, and phytosterols	450 or 900 mg/kg, 4 weeks	Rats with high-fat diet	Hypolipidemic action	[28]
Clinical model					
Pulp	MUFAs	136 g/p.o., 5 weeks	Overweight or obese humans	Hypolipidemic action	[103]
Pulp	MUFAs	75% avocado-derived MUFAs/p.o., 4 weeks	Dyslipidemia patients	Hypolipidemic action	[104,105]
Pulp	MUFAs	75% avocado-derived MUFAs/p.o., 2 weeks	Healthy humans	Hypolipidemic action	[106]
Pulp	No data	1 avocado (200 g)/p.o., 6 weeks	Overweight or obese humans	Anti-obesity action	[107]
Pulp	MUFAs	1 avocado or more, 1 week, for 4 years	Healthy humans	Reducing risk of coronary heart disease and developing of CVDs	[14]
Pulp	Persenone C	500 µg/mL	Healthy humans	Ant-platelet action	[55]

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
