# Peer review of "The Pulp, Peel, Seed, and Food Products of *Persea americana* as Sources of Bioactive Phytochemicals with Cardioprotective Properties: A Review"

_ijms, 2024, doi:10.3390/ijms252413622_

Round 1

Reviewer 1 Report

Comments and Suggestions for Authors The review "The pulp, peel, seed and food products of Persea americana as sources of bioactive phytochemicals with cardioprotective properties: a review"  describes the composition and potential cardioprotective roles of Persea americana. Recent papers have been published on the bioactive compounds of Avocado    I would recommend to revise the paper to improve reading   -Separate tables from figures  Table 1 and Figure 1 are best separated by text. Figure 2 and  table 3 are best separated by text.   -lines 273-274 . Add reference: In addition, avocado and its food products are good sources  of compounds with known cardioprotective properties, such as potassium, magnesium, and vitamins such as vitamin A.   -lines 284-288 The authors should describe better the role of oleic acid. Which acyltransferase is involved?    -the part titled "Bioavailability of avocado chemical compounds" contains data only on carotenoids. What about other bioactive molecules?   -lines 299 - 300 what is the meaning of the sentence Moreover, Tan et al. [28] noted that the ingestion of avocado oil and simvastatin causes a decrease in the atherogenic index, thus reducing the risk of atherosclerosis.   -lines 310-312 What is the levels of  phytosterols in avocado pulp? the effect of  phytosterols on LDL-cholesterol is concentration-dependent.    -The authors describe the composition of different products (pulp, seed etc,..) A table summarizing non only the list of molecules but also the levels of the main bioactive compound should be included    -What about the techniques used to separate the oil or seed? solvent extraction or other methods modulate chemical composition of the products?   -Table 3 Describe better the studies   -Is an in vitro in vivo study? Pulp Persenone C 500 µg/mL Healthy humans Ant-platelet action [55]   Pulp No data 1 avocado or more?  1 week Healthy humans Reducing risk of coronary heart disease and developing of CVDs [14] 

Author Response

The review "The pulp, peel, seed and food products of Persea americana as sources of bioactive phytochemicals with cardioprotective properties: a review"  describes the composition and potential cardioprotective roles of Persea americana. Recent papers have been published on the bioactive compounds of Avocado    I would recommend to revise the paper to improve reading  

Thank you for reviewing the manuscript and providing such helpful comments. All of them have been taken into consideration when revising the manuscript.

-Separate tables from figures  Table 1 and Figure 1 are best separated by text. Figure 2 and  table 3 are best separated by text.  

Response: I have corrected.

-lines 273-274 . Add reference: In addition, avocado and its food products are good sources  of compounds with known cardioprotective properties, such as potassium, magnesium, and vitamins such as vitamin A.

Response: I have added references: [3,17,38].

 -lines 284-288 The authors should describe better the role of oleic acid. Which acyltransferase is involved?   

Response: I have added more information about it: “Oleic acid can decrease LDL-cholesterol by increasing acyl-CoA: cholesterol acyltransferase (a liver enzyme that catalyzes the formation of cholesterol esters from cholesterol) activity, thus increasing the synthesis of cholesterol esters; these stimulate the action of LDL receptors, favoring LDL uptake and reducing its presence in plasma [77]. Other studies showed that decreasing plasma triglyceride content and increasing that of HDL-cholesterol may favor the lipase hydrolysis of long-chain fatty acids in the triglycerides, which are incorporated into HDL particles [78]. In addition, oleic acid may reduce endogenous total cholesterol synthesis, and decrease LDL-cholesterol and total cholesterol levels [77].

-the part titled "Bioavailability of avocado chemical compounds" contains data only on carotenoids. What about other bioactive molecules?

Response: There are only few information about the bioavailability of the compounds in avocado alone (especially carotenoids), others have examined their absorption when used in combination with other foods and supplements [4,44]. Interestingly, avocado can act as a “nutrient booster” by improving the absorption of fat-soluble nutrients found in foods. For example, avocado has been found to improve carotenoid absorption consumed with carotene-rich tomato source or carrots in healthy subjects [45]. Elsewhere, a study of 11 healthy adults found the absorption of β-carotene and lycopene from salsa to increase when avocado (24 g) was added (150 g) [44]. The authors report higher circulating levels of β-carotene (2.6 x) and lycopene (4.4 x) in the salsa with avocado group compared to the without avocado group; they attribute this process to the high MUFA content of avocado. Furthermore, the highly-bioavailable lutein from avocado also helps protect low-density lipoprotein (LDL) cholesterol from oxidation and can inhibit atherosclerosis, which can benefit cardiovascular health. A higher lutein intake is also associated with lowered risk of stroke and coronary heart disease [46,47].

-lines 299 - 300 what is the meaning of the sentence Moreover, Tan et al. [28] noted that the ingestion of avocado oil and simvastatin causes a decrease in the atherogenic index, thus reducing the risk of atherosclerosis.  

Response: I have added more information about it: „Authors studied the hypocholesterolaemic effect of virgin avocado oil (400 and 900 mg/kg body weight) using diet-induced hypercholesterolaemia rats. Rats were fed high-cholesterol diet for 4 weeks to induce hypercholesterolaemia. At the end of the experiment, the LDL-cholesterol and triglyceride  levels were significantly reduced, while the HDL-cholesterol level was significantly increased in high-dose avocado oil (900 mg/kg body weight per day) when compared with their respective baseline values.”

-lines 310-312 What is the levels of  phytosterols in avocado pulp? the effect of  phytosterols on LDL-cholesterol is concentration-dependent. 

Response: I have added this information – chapter 2: „Phytocehmical characteristics….”: „Six phytosterols have been detected in avocado pulp: β-sitosterol, stigmasterol, campesterol, cyclosterol, avenasterol, and stanol; the most abundant of these is β-sitosterol (57 – 76 mg/100 g) [3]. Avocado also contains phytostanols, the main one being cycloarthenol. Avaocado oil is also the source of phytosterols and phytostanols such as lanosterol, campestanol, Δ5-avenasterol, Δ7-sitosterol, cycloeucalenol, citrostadienol, and 24-methylenecycloartanol [15].” 

 -The authors describe the composition of different products (pulp, seed etc,..) A table summarizing non only the list of molecules but also the levels of the main bioactive compound should be included 

Response: There are only few information about concentration of different bioactive compounds. I have added this information in Table 1 and in the text of manuscript – chapter 2.

-What about the techniques used to separate the oil or seed? solvent extraction or other methods modulate chemical composition of the products?  

Response: I have added more information about it. For example, “Avocado oil is mainly obtained from the pulp by mechanical techniques such as cold pressing, or chemical solvation with organic solvents. Avocado oil contains high levels of MUFA, constituting 37 – 86% of total fatty acid content, with the main form being oleic acid; however, there are no internationally-defined parameters for avocado oil content. Various authors have noted that oil quality and yield depends on the cultivar, quality and maturity of the fruits, as well as the extraction method and conditions, such as the chosen solvent, temperature, pH, and added enzymes [8]. For example, the quality parameters, including amount of oleic acid, content of carotenoids, phenolic compounds, and antioxidant properties have shown better results when the pulp is dried at 60C under vacuum, and the extraction is performed by the Soxhlet method. However, the bioactive compounds were best preserved when the avocado pulp was dried at 60C with air ventilation and mechanical pressing [8].

The resulting oil is used both in cosmetics and food production [25]. Extra-virgin avocado oil, for example, is a viscous edible oil with a dark green color due to its carotenoid and chlorophyll contents. It has a mild taste and it is typically extracted from avocado fruit by cold pressing [25].

Avocado oil may also be extracted from the seed and skin. The edible oil obtained from avocado seeds has its own flavor, color and texture, and interestingly, crude avocado oil has a similar viscosity to extra-virgin avocado oil. This may not surprising considering that refined avocado oil is obtained mainly from crude oil. Crude oil is light in color, odorless, free of waxes, and low in free fatty acids [25]. A more detailed physiochemical characterization of the oils from different varieties and origin of avocado is given in a review by Flores et al. [8], who also note that pulp oil is has higher proportions of MUFA than the seed oil, while the seed oil has higher levels of PUFA than the pulp oil [26].”

-Table 3 Describe better the studies   -Is an in vitro in vivo study? Pulp Persenone C 500 µg/mL Healthy humans Ant-platelet action [55]   Pulp No data 1 avocado or more?  1 week Healthy humans Reducing risk of coronary heart disease and developing of CVDs [14] 

Response: I have added information about experimental models (animal models and clinical models) – table 3. In addition, I have added more information about experiment described in paper [14] and [55].

Reviewer 2 Report

Comments and Suggestions for Authors

This article introduces Avocado (Persea americana) which is a nutrient-dense fruit with a creamy, smooth-textured pulp and a thick, bumpy skin. It contains bioactive compounds beneficial for cardiovascular health. Avocado is rich in monounsaturated fatty acids (MUFAs), dietary fiber, potassium, magnesium, and various vitamins. It has a high fat content, predominantly oleic acid. The pulp, peel, and seed contain phenolic compounds, fatty alcohols, and carotenoids, which have antioxidant, anti-inflammatory, and cardioprotective properties. Regular avocado consumption is linked to improved cardiovascular health, reduced cholesterol levels, and lower risk of coronary heart disease and other cardiovascular diseases. However, there are still some questions about the content of the article.

What is the yield of the waste parts, and is it convenient to collect them to support their application in the food and cosmetic industries?

The grammar of this article is generally correct, but there are some areas that can be improved for better fluency and clarity. Here are some suggestions:

Sentence Structure: Some sentences are long and complex, and can be broken into shorter sentences.

Line14: “These have been obtained from the pulp, the peel, and the seed.” Improved to “These compounds have been obtained from the pulp, peel, and seed.”

Ensure consistent use of terms. For example, “Persea americana” and “P. americana” should be explained at first mention and used consistently thereafter.

There are some spelling errors such as Line127, “source”. Maybe you should use “sources”.

Remove unnecessary repetitive information. For example: Line23: “The avocado itself grows in a warm temperate climate. It is believed to have been first cultivated in Mexico…” could be improved to : “Avocado grows in a warm temperate climate and is believed to have been first cultivated in Mexico…”

Line 375, conclusion should be the 6th part.

Author Response

This article introduces Avocado (Persea americana) which is a nutrient-dense fruit with a creamy, smooth-textured pulp and a thick, bumpy skin. It contains bioactive compounds beneficial for cardiovascular health. Avocado is rich in monounsaturated fatty acids (MUFAs), dietary fiber, potassium, magnesium, and various vitamins. It has a high fat content, predominantly oleic acid. The pulp, peel, and seed contain phenolic compounds, fatty alcohols, and carotenoids, which have antioxidant, anti-inflammatory, and cardioprotective properties. Regular avocado consumption is linked to improved cardiovascular health, reduced cholesterol levels, and lower risk of coronary heart disease and other cardiovascular diseases. However, there are still some questions about the content of the article.

Thank you for reviewing the manuscript and providing such helpful comments. All of them have been taken into consideration when revising the manuscript.

What is the yield of the waste parts, and is it convenient to collect them to support their application in the food and cosmetic industries?

Response: I have added short information about it (the chapter – Introduction): “Production and consumption of avocado generate large quantities of by-products, including peel, seeds, and defatted pulp, which account for approximately 30-45% of fruit weight, but for example the seed may be used in the food and cosmetic industries: it can be found in flour, orange dye, skin products, and soap [6].

The grammar of this article is generally correct, but there are some areas that can be improved for better fluency and clarity. Here are some suggestions:

Sentence Structure: Some sentences are long and complex, and can be broken into shorter sentences.

Line14: “These have been obtained from the pulp, the peel, and the seed.” Improved to “These compounds have been obtained from the pulp, peel, and seed.”

Response: I have corrected. Now, it is: „These compounds have been obtained from the pulp, peel, and seed.”

Ensure consistent use of terms. For example, “Persea americana” and “P. americana” should be explained at first mention and used consistently thereafter.

Response: I have corrected.

There are some spelling errors such as Line127, “source”. Maybe you should use “sources”.

Response: I have corrected. Now, it is: „sources”.

Remove unnecessary repetitive information. For example: Line23: “The avocado itself grows in a warm temperate climate. It is believed to have been first cultivated in Mexico…” could be improved to : “Avocado grows in a warm temperate climate and is believed to have been first cultivated in Mexico…”

Response: I have corrected. Now, it is: „Avocado grows in a warm temperate climate and is believed to have been first culti-vated in Mexico…”.

Line 375, conclusion should be the 6th part.

Response: I have corrected. Now, it is:„6. Conclusion”.